# Will Bid/No-Bid Decision Factors for Construction Projects Be Different in Economic Downturns? A Chinese Study

**Jinxiu Wang [1], Lin Wang [1], Kunhui Ye [2,*] and Yongwei Shan [3]**

[1]  School of Management Science and Real Estate, Chongqing Univ., No. 83 Shabei St., Shapingba District, Chongqing 400045, China; wangjx81911@126.com (J.W.); wangidill@163.com (L.W.)

[2]  International Research Center for Sustainable Built Environment, Chongqing Univ., No. 83 Shabei St., Shapingba District, Chongqing 400045, China

[3]  School of Civil and Environmental Engineering, Oklahoma State Univ., 248 Engineering North, Stillwater, OK 74078, USA; yongwei.shan@okstate.edu

*  Correspondence: Kunhui.YE@gmail.com

**Abstract:** Whether to bid on construction work is a traditional question that is often challenged in different economic situations. What will the decision factors be when contractors have to struggle to achieve growth in economic downturns? Since previous studies have not widely agreed with each other on this issue, this study aims to examine China's construction engineering contractors in recent "bad" years. The research findings reveal that the cluster related to expected profitability, including terms of payment, reputation of the client regarding his/her commitment to making timely payments, original price estimated by the client, profit track record for similar projects, and contract type, were the most significant determinants. The bid/no-bid decision-making process comprises three modules, including the contractor's strategies, the contractor's competency for a project, and expected competition. These results suggest that competing for survival is the dominant position of contractors in making bid/no-bid decisions in economic downturns.

**Keywords:** bidding decision; construction projects; economic downturns; hierarchical cluster analysis; China

## 1. Introduction

Finding a project to bid on is essential for construction engineering contractors to survive market competition [1]. While the construction market offers contractors considerable opportunities to bid, it requires contractors to carefully select construction projects that are most suitable to them. However, contractors' decision-making under a bid/no-bid (BNB) scenario is an outstanding managerial problem that has not been adequately addressed in practice or in academia [2–4]. Efficient BNB decision helps contractors earn projects and market position. In contrast, poorly thought-out decisions will plague a contractor's profitability and reputation [5]. Due to the complexity of bidding, contractors' BNB decisions have recently attracted close attention in the discipline of construction engineering management. A consensus reached in prior research is that such decisions are puzzling and time-consuming; a handful of factors, such as the macro-economy, client identity, and workload, also deserve attention, and simplifying these factors into an effective decision point is necessary for businesses to stay competitive in the current age [6–9].

Factors affecting contractors' BNB decisions are not universal but are highly related to the economic environment. These factors include employment, income, inflation, and productivity [10]. The economic environment is linked to the fluctuation of economic growth, and each economic entity

has its own development cycle for boom and bust [11]. Rapid economic growth with a low inventory level and high confidence in a boom is a trigger for societal investment [12], and under such conditions, more construction projects will be demanded. In these good times, a smart contractor knows to expand his or her business to properly reap the profits [13]. However, contractors must operate under circumstances in which the rate of economic growth drops significantly. According to the accelerator effect theory, a drop in the GDP growth rate may cause a larger drop in investment [14], which means that new construction projects will be less frequently launched. In economic downturns, contractors are faced with the challenges of shrinking demand and stronger pressure from business competition [15,16]. Uncompetitive firms will be outperformed by their competitors and exit the market. Therefore, contractors should be prudent when selecting new projects to bid on and attempt to avoid pitfalls in BNB decisions during economic downturns [17].

One of the largest developing countries in the world, China, has been one of the fastest growing economies for decades. However, recent years have witnessed China's economy suffer from a downturn after the global financial crisis in 2008. The growth rate of China's GDP fell from 14.2% in 2007 to 6.1% in 2019 [18,19]. In parallel with the wave of economic development, the total added value in the Chinese construction industry reached 16.2% in 2007 but then fell to 4.5% in 2018 [20]. The Chinese government has been making numerous efforts to stimulate economic growth to guarantee the prosperity of the construction industry during economic downturn periods. Nevertheless, the growth of the construction industry may not sustain the same growth as the economy. The increase in the intensity of competition caused by economic downturns drives many uncompetitive firms to abandon their businesses [21]. Consequently, the growth of construction enterprises becomes stagnant [21]. In such bad times, contractors must guarantee effectiveness and efficiency in their BNB decisions. It is thus considered necessary to explore contractors' BNB decision-making factors.

Despite the richness of previous studies, researchers have not widely agreed with one another on contractors' BNB decisions during economic downturns. For instance, the top factors identified in economic downturns, including contractors' abilities to undertake construction projects [4,6,7,10,22] and pursue profitability [3,23], change between different countries. Although the rise of Chinese contractors in the international construction industry has been fueling researchers' interest over the past few decades [24], previous studies devoted to the theme of Chinese contractors' BNB decisions are scarce. It is thus necessary to re-examine and elaborate the factors of BNB decisions in China's recent economic downturns and to provide evidence to aid academic debates in the related research area. Moreover, identification of these factors can help Chinese contractors develop more effective competition strategies and make correct decisions on whether or not to bid during "bad" periods.

In this study, a set of tentative factors for contractors' BNB decisions was derived via a literature review and an interview. A survey was conducted to collect relevant practitioners' opinions on the importance level for each factor. The critical factors were identified through a rank analysis, and the identified factors were grouped using a hierarchical cluster analysis (HCA) approach. The clusters and the factors included under each cluster were elaborated. The research findings offer some suggestions that may help contractors fortify their competitive advantages in economic downturns. Despite using China as a case, the present results are expected to shed light on the same subjects in construction markets in other countries.

## 2. Literature Review

### 2.1. Attributes of Contractors' BNB Decisions

Contractors' BNB decisions are complicated. A typical example is the study by Jarkas et al. [10], which considers the top factors (i.e., the contractor's experience with the owner, the contractor's need for work, the contractor's current workload, the contractor's experience with the project, and the project size) in Qatar to be the comprehensive ability of the contractor to undertake the project.

Jarkas et al.'s findings suggest that contractors' BNB decision factors range from their current resources to the accumulation of experience.

The complexity of contractors' BNB decisions is reflective of some competing factors identified in previous studies. For instance, Cheng et al. [23] stated that the top factor in Vietnam was expected profitability. In contrast, in the United States, Ahmad and Minkarah [6] found that the top factors for contractors' BNB decisions were a contractor's need for work, rather than expected profit. This finding was surprisingly echoed by Shash [4] in a similar examination of the UK's construction industry. Furthermore, Bageis and Fortune [7] revealed that contractors in Saudi Arabia have to manage aspects related to the financial capacity of the owner, the project's cash flow, the contractor's ability to do the project, the availability of the required money, and start-up capital. It is thus reasonable to infer that the dominant BNB decision factors relate to whether the funds of the contractor can meet the requirements of the expected cash flow of construction projects. Similarly, El-Mashaleh [3] demonstrated that the factors in Jordan were the owner's reputation for making timely payments, the financial capability of the owner, and the identity of the owner. However, Wanous et al. [22] claimed that the top factors in Syria were a contractor's bidding qualifications, as well as owners, project size, and the availability of time to tender.

### 2.2. Contractors' BNB Decision Factors

A tentative list of 31 factors frequently cited in the related literature is shown in Table 1. These factors are restructured in line with Ahmad's [25] proposal, which offers generic criteria for categorizing the factors into three groups: project-related, firm-related, and market-related factors. As Table 1 indicates, the diversity of these factors suggests that contractors' BNB decisions are not uniform. First, project-to-bid is one of the most crucial decision factors. A contractor's determination to bid is likely due to the attractiveness of a project's location [4], type [4], size [26], complexity [27], and profit track record for similar projects [28]. Second, clients' requirements are another aspect that contractors should pay attention to. Contractors have to evaluate, in detail, clients' reputations and financial resources [3], the contract type [10], the criteria for bid selection [8], the original price estimated by the client [7], the project duration [4], required subcontractors [3], the quality of bidding documents (e.g., drawings and specifications) [3], security requirements (e.g., bid security and performance security) [3], terms of payment [26], and time for bid preparation [8]. Third, contractors' situations are not exclusive and include the needs of the work (i.e., labor force, materials, equipment, and funds) [8], current workload [2], contracting experience [3], financial situation [26], relationship with the client [29], and compliance with strategies [23]. Fourth, competitor analysis is a key element of BNB decisions. The elements of competitor analysis include the availability of other projects [2], the competence of the expected competitors [23], the current workloads of the expected significant competitors [3], and the number of expected competitors [4].

**Table 1.** Factors affecting contractors' BNB decisions.

| Basic Category | Code | Factor | Reference |
|---|---|---|---|
| Project related | SF1 | Amount of work that the client carries out regularly | [26] |
| | SF2 | Required subcontractors | [3] |
| | SF3 | The complexity of the project | [27] |
| | SF4 | Contract type | [10] |
| | SF5 | Criteria of bid selection | [8] |
| | SF6 | Influence of the client in making recommendations in the construction market | [3] |
| | SF7 | Needs of the work (i.e., labor force, material, equipment, and funds) | [8] |
| | SF8 | Original price estimated by the client | [7] |
| | SF9 | Profit track record for similar projects | [28] |
| | SF10 | Project duration | [4] |
| | SF11 | Project location | [4] |
| | SF12 | Project size | [26] |
| | SF13 | Project type | [4] |
| | SF14 | Quality of bidding documents (e.g., drawings and specifications) | [3] |
| | SF15 | Reputation of the client regarding his/her commitment to making timely payments | [3] |
| | SF16 | Security requirements (e.g., bid security and performance security) | [3] |
| | SF17 | Terms of payment | [26] |
| | SF18 | Time for bid preparation | [8] |
| Firm related | SF19 | Compliance with business strategy | [23] |
| | SF20 | Current workload of projects | [2] |
| | SF21 | The current workload in bid preparation | [2] |
| | SF22 | Establishing a long relationship with the client | [29] |
| | SF23 | Experience of the firm with similar projects | [3] |
| | SF24 | Improving the experience of the firm's personnel | [3] |
| | SF25 | Increasing the possibility of entering new markets | [3] |
| | SF26 | Promoting the reputation of the firm | [3] |
| | SF27 | The current financial situation of the firm | [26] |
| Market related | SF28 | Availability of other projects | [2] |
| | SF29 | Competence of the expected competitors | [23] |
| | SF30 | Current workloads of the expected significant competitors | [3] |
| | SF31 | Number of expected competitors | [4] |

## 2.3. Contractors' BNB Decision Factors during Economic Downturns

The difficulties in business operations during economic downturns have compelled many researchers to examine BNB decision factors, as shown in Table 2. Different economic situations require businesses to adjust their competition strategies to improve their competitiveness, especially when an economic downturn is approaching. Basically, an economic downturn refers to the period when the economic growth rate decreases. The characteristics of an economic downturn include the low confidence of investors and a smaller amount of societal investment [30]. This decrease in societal investment means that fewer new construction projects are available for contractors to bid on. Furthermore, according to the market supply and demand theory, business competition in the construction market will intensify, prices for transactions will decrease, and profit will decrease in economic downturns. In these challenging times, it is crucial for businesses to moderate the relationship between input and output to maximize their profits with limited resources, including labor and financial capacity. Likewise, in economic downturns, contractors should reduce their expectations of new contracts and cautiously choose projects to bid on in order to maximize their profits.

**Table 2.** The economic downturn extent in previous studies on contractors' bid/no-bid decision factors.

| Data Collection Year | Country/Region | GDP Growth Rate | Reference |
|---|---|---|---|
| 1988 | United States | Fell from 7.26% in 1984 to −0.07% in 1991 | [7] |
| 1990 | UK | Fell from 5.75% in 1988 to −1.09% in 1991 | [4] |
| 1999 | Syria | Fell from 13.47% in 1992 to −3.55% in 1999 | [1] |
| 2009 | Saudi Arabian | Fell from 11.24% in 2003 to −2.06% in 2009 | [8] |
| 2009 | Vietnam | Fell from 7.55% in 2005 to 5.40% in 2009 | [23] |
| 2013 | Jordan | Fell from 8.57% in 2004 to 2.0% in 2016 | [3] |
| 2014 | Qatar | Fell from 19.59% in 2010 to 2.22% in 2016 | [11] |

Note: GDP data from the World Bank website.

## 3. Materials and Methods

### 3.1. Research Method

A quantitative method was adopted to elaborate the key factors determining contractors' BNB decisions. First, a set of preliminary factors was derived via a literature review (Table 1) and refined through an interview with professionals. Second, a survey was conducted to collect professional opinions on the importance of the preliminary factors. Third, the collected data were analyzed to examine the importance level for each factor. Fourth, the critical factors were identified through a rank analysis. Last, the identified essential factors were grouped using the HCA approach.

Five senior professionals (i.e., a general manager from Chongqing, two project managers from Beijing and Guangdong, and two department managers from Shanghai and Shandong) were invited by convenience to make comments on the tentative factors. As a result, local industry policies (SF32) and the local social environment (SF33) (which are significant influential factors that affect a contractor's ability to win a contract and also affect project performance) were added to the list of factors. Consequently, 33 factors were derived for the questionnaire survey in China.

### 3.2. Data Collection

A questionnaire was designed with a web tool for data collection. The questionnaire was composed of three parts. The first part describes an overview of the survey. The second part is intended to collect demographic data concerning the respondents' position, professional background, years of work, and their firm's professional category. In the third part, the respondents are invited to mark the importance level of the 33 factors. A 1 to 7 Likert scale is used for this measurement, where one represents exceptionally unimportant, four represents neutral, and seven represents extremely important. This survey was designed so that respondents would not be able to successfully submit their answers if any item in the questionnaire was not answered, as a warning message would be generated.

The following formula was adopted to determine the sample size [7]:

$$n = n1/(1 + n1/N) \tag{1}$$

where n = the sample size; $n1 = S^2/V^2$; N = the total population (the number of construction firms = 95,400 [20]); V = the standard error of the sampling distribution = 0.05; $S^2 = P (1 − P) = 0.5 \times (1 − 0.5) = 0.25$ (S = the maximum standard deviation in the population elements, the total error = 0.1, the confidence level is 95%, and P = the proportion of population elements that belong to the defined class).

The value of the predefined variables was substituted into the formula mentioned above, and then the sample size n = 100 was derived.

The online questionnaire was distributed electronically to practitioners working for general contractors across China in October 2018. The total number of invited respondents was 500, from which 112 returned responses (a 22.4% response rate). Among all the returned questionnaires, 109 were found to be valid, and the others were not taken seriously (such as those with the same rating for all items). The valid respondents included 18 top executives, 32 project managers, and 59 department managers. Moreover, all respondents had more than five years of work experience, and over 65.75% of

them had over 10 years of experience. The position and years of work experience were used to judge the validity of the samples. It is essential to ensure that the participants have sufficient knowledge of BNB decisions and are involved in the decision-making process. For each type of construction, 72.60% of the respondents were from commercial building construction, 12.33% from heavy construction, 10.96% from local public construction, 2.74% from industrial construction contractors, and 1.37% were utility contractors. In terms of the distribution of the respondents' geographical locations, the respondents were distributed over 20 provinces and cities, including Beijing (6), Fujian (7), Guangdong (8), Jiangsu (12), Shandong (8), Shanghai (6), Zhejiang (9), Anhui (4), Guangxi (2), Guizhou (4), Henan (5), Hunan (4), Liaoning (2), Neimenggu (2), Gansu (2), Shaanxi (4), Shanxi (3), Sichuan (5), Yunnan (3), and Chongqing (13). It is difficult to assess the extent to which the samples represent the entire Chinese construction industry. However, the composition of the respondents, to some extent, helps avoid biases.

*3.3. Data Analysis*

Before further analysis, it is necessary to examine the internal consistency of the data regarding the importance levels of the factors. With the assistance of Statistical Product and Service Solutions (SPSS 19.0 software developed by IBM and headquartered in Armonk, NY, USA), the computed coefficient of the internal consistency test, Cronbach's alpha, was 0.891 (>0.7), showing a high internal consistency of the factors.

Based on the mean value of each factor, all factors were ranked to identify the important ones. Factors with mean values greater than four are considered to be important factors.

Since some of these critical factors are similar, it is necessary to group critical factors to obtain a more precise structure for efficient decision-making. To obtain the desired result, HCA was utilized to group factors. This method is a common statistical technique for identifying and presenting the similarities between multiple attributes in the form of dendrograms, and clusters are identified according to the needs of the research [31]. The investigated opinions on the importance level of the critical factors of each dimension (i.e., project-related, firm-related, and market-related factors) were fed into SPSS 19.0 software to conduct the HCA. Before the HCA, the internal consistency of project-related, firm-related, and market-related data was tested separately.

An importance index was developed and employed to examine the clusters [32]. The mean value of the importance of the factors, including the importance level per cluster, was calculated using the following formula:

$$\text{Importance index } = \frac{1}{n} \sum_{1}^{n} x_i \times \frac{100}{7}$$

where n = the total number of factors included in the cluster, and $x_i$ = the selected factor i's mean value.

**4. Results and Findings**

The ranking results are presented in Table 3. Of all the factors, only time for bid preparation (SF18) and current workloads of the expected significant competitors (SF30) had a mean value less than or equal to four, which means that SF30 and SF4 were not significant factors. As a result, 31 factors were rated as essential factors for contractors' BNB decisions. The terms of payment (SF17), reputation of the client regarding his commitment to making timely payments (SF15), original price estimated by the client (SF8), profit track record for similar projects (SF9), and contract type (SF4) were the top five factors and are related to project profitability.

**Table 3.** Rank of factors affecting contractors' BNB decisions.

| Rank | Code | Mean |
|------|------|------|
| 1 | SF17 $^\Delta$ | 6.29 |
| 2 | SF15 $^\Delta$ | 6.23 |
| 3 | SF8 $^\Delta$ | 5.90 |
| 4 | SF9 $^\Delta$ | 5.85 |
| 5 | SF4 $^\Delta$ | 5.75 |
| 6 | SF22 $^\blacklozenge$ | 5.66 |
| 7 | SF13 $^\Delta$ | 5.58 |
| 8 | SF26 $^\blacklozenge$ | 5.48 |
| 9 | SF25 $^\blacklozenge$ | 5.34 |
| 10 | SF12 $^\Delta$ | 5.29 |
| 11 | SF7 $^\Delta$ | 5.19 |
| 12 | SF20 $^\blacklozenge$ | 5.19 |
| 13 | SF27 $^\blacklozenge$ | 5.16 |
| 14 | SF5 $^\Delta$ | 5.14 |
| 15 | SF6 $^\Delta$ | 5.14 |
| 16 | SF19 $^\blacklozenge$ | 5.12 |
| 17 | SF10 $^\Delta$ | 5.11 |
| 18 | SF3 $^\Delta$ | 4.99 |
| 19 | SF33 * | 4.88 |
| 20 | SF16 $^\Delta$ | 4.84 |
| 21 | SF23 $^\blacklozenge$ | 4.84 |
| 22 | SF32 * | 4.82 |
| 23 | SF14 $^\Delta$ | 4.78 |
| 24 | SF29 * | 4.67 |
| 25 | SF24 $^\blacklozenge$ | 4.62 |
| 26 | SF33 * | 4.60 |
| 27 | SF31 * | 4.49 |
| 28 | SF11 $^\Delta$ | 4.38 |
| 29 | SF21 $^\blacklozenge$ | 4.36 |
| 30 | SF2 $^\Delta$ | 4.29 |
| 31 | SF1 $^\Delta$ | 4.08 |
| 32 | SF30 * | 4.00 |
| 33 | SF18 $^\Delta$ | 3.82 |

Note: $^\Delta$ Project-related factors; $^\blacklozenge$ Firm-related factors; * Market-related factors.

Figure 1, Figure 2, and Figure 3 show the cluster results for project-related, firm-related, and market-related factors, respectively. The intuition based on these figures is that each dimension has two clusters, which are the lowest clusters of the HCA. In total, six clusters were determined. Each cluster shown in Table 4 was given a name to reflect the attributes of the clustered factors. The names of the six clusters are expected profitability, project requirements, organizational goals, organizational resources, expected competition, and the local business environment. By using the importance index, the rank of each cluster was generated, as shown in Table 4. Expected profitability and organizational goals comprise the top clusters, and expected competition ranks at the bottom.

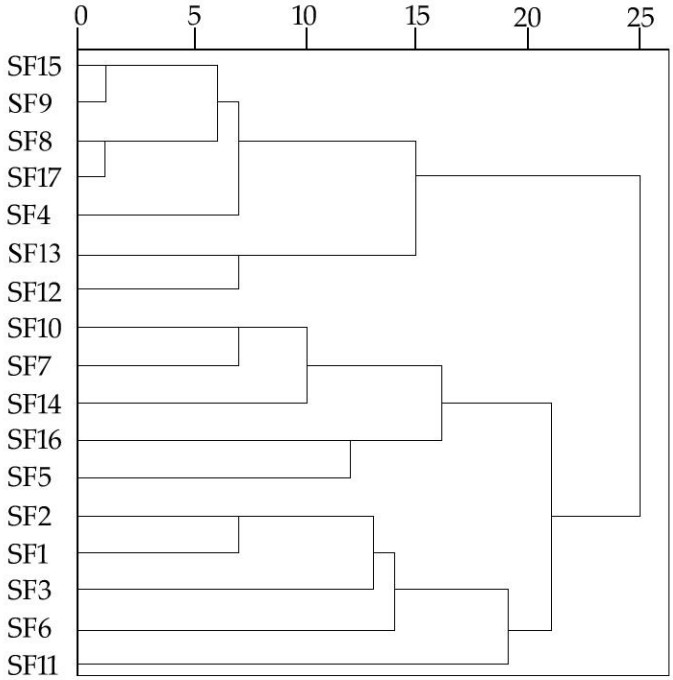

**Figure 1.** The cluster results of important project-related factors.

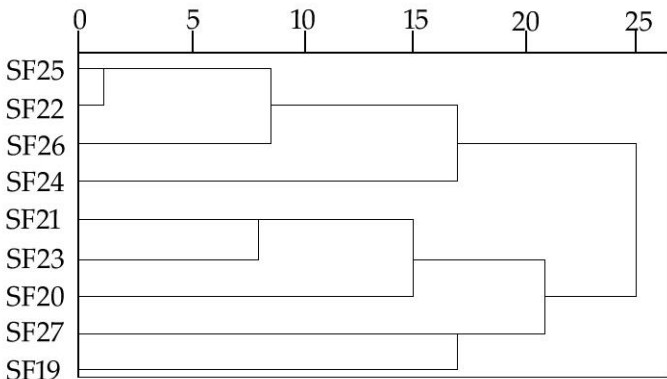

**Figure 2.** The cluster results of important firm-related factors.

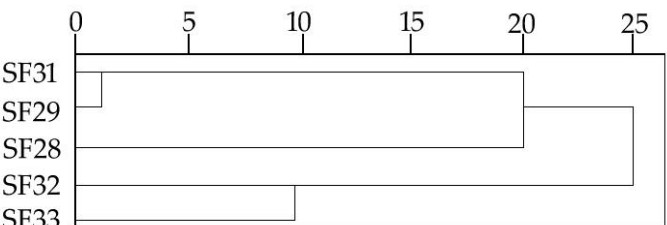

**Figure 3.** The cluster results of important market-related factors.

**Table 4.** Results of hierarchical cluster analysis (HCA) of the critical factors.

| Basic Category | Cronbach's Alpha | Cluster | Importance Index | Rank | Factor Included in the Cluster |
|---|---|---|---|---|---|
| Project related | 0.801 | Cluster 1: expected profitability | 83.45 | 1 | SF4<br>SF8<br>SF9<br>SF12<br>SF13<br>SF15<br>SF17 |
| | | Cluster 2: project requirements | 68.49 | 5 | SF1<br>SF2<br>SF3<br>SF5<br>SF6<br>SF7<br>SF10<br>SF11<br>SF14<br>SF16 |
| Firm-related | 0.826 | Cluster 3: organizational goals | 75.36 | 2 | SF22<br>SF24<br>SF25<br>SF26 |
| | | Cluster 4: organizational resources | 70.49 | 3 | SF19<br>SF20<br>SF21<br>SF23<br>SF27 |
| Market-related | 0.746 | Cluster 5: expected competition | 65.52 | 6 | SF28<br>SF29<br>SF31 |
| | | Cluster 6: local business environment | 69.29 | 4 | SF32<br>SF33 |

## 5. Discussion

The clusters identified above are discussed following their ranking in descending order, as shown in Table 4. The differences and implications of the clusters and the included variables in each cluster are presented below.

### 5.1. Cluster 1: Expected Profitability

As shown in Table 4, all seven items contained in cluster 1 are related to expected profitability. Profit is a core source of a firm's development. The identification of this factor concurs with previous studies in the construction industry. As pointed out by Wanous, Boussabaine, and Lewis [22], most bidding models were devised to adhere to the principle of profit maximization. In reality, contractors are apt to evaluate the expected profitability of new projects in the bidding phase and consider it to be a core criterion when deciding whether to bid [6]. The amount of profit and payment efficiency are two significant aspects of profitability [33]. Various factors, including contract type (SF4), original price estimated by the client (SF8), profit track record for similar projects (SF9), project size (SF12), and project type (SF13), can be used to estimate the amount of profit, and factors including the reputation of the client regarding his commitment to making timely payments (SF15) and terms of payment (SF17) are related to payment efficiency.

Table 4 shows that expected profitability is the key focus of contractors' BNB decisions during economic downturns, which is different from most of the previous studies. One of the main reasons is that the Chinese construction market is unique, and the generally low-profit margins of the industry [20] cause contractors to have distinctive behaviors. In addition to the amount of profit, contractors in "bad" years pay more attention to payment efficiency. Table 3 shows that the two factors involved in payment efficiency can be ranked as the top two factors. This result is in line with some investigations in the UK [9], northern Cyprus and Turkey [26], and Jordan [3]. Given the time-value of money, low payment efficiency reduces the amount of profit, especially when a construction project lasts 1–2 years or even longer. Chinese contractors are reluctant to accept when the amount of uncollected progress payments

from the client and payment lag time keep growing, especially in the housing construction market [21]. Anecdotal evidence shows that some projects in China were denied by prudent contractors because of their undesirable payment terms, regardless of the size of the expected profit. This strategy is widely accepted, especially among small and medium-sized contractors, which are more likely to be dragged down by considerable investment in a project with low payment efficiency.

### 5.2. Cluster 3: Organizational Goals

The four factors in cluster 3 are related to organizational goals. Organizational goals are derived from decision makers' motivation under the constraint sets. A company must set goals under the leadership of upper management to survive and develop based on the internal and external conditions [34]. Likewise, construction contractors should engage in appropriate strategies and develop similar goals based on their actual individual situations. In addition to the short-term goal of maximizing expected profitability, a contractor may have alternative goals to develop sustainable competitiveness, especially during economic downturns. It is practical for a contractor to choose a construction project that can meet its development goals. Otherwise, the contractor's competitiveness can be compromised in the rapidly changing market.

The four factors of organizational goals involve four types of contractor development goals. Establishing long relationships with the client (SF22) and promoting the reputation of the firm (SF26) are two activities of corporate social responsibility (CRS) defined by Petrovic-Lazarevic [35]. SF22, SF25 (increasing the possibility of entering new markets), and SF26 are included in the top ten factors that determine contractors' BNB decisions in China. These three factors can offer significant competitiveness for the contractor to win new business; in particular, obtaining enough project contracts is vital in the Chinese construction industry [36]. Firstly, a good reputation (SF26) may be helpful in raising a contractor's profile during prequalification [37], as well as raising his or her bargaining power with clients. Improving the reputation of the firm is one of the most effective development strategies for contractors, especially in this intensely competitive market. SF26 was also studied as an important subject in BNB decisions in Jordan by El-Mashaleh [3]. Secondly, entering new markets (SF25) also becomes an important development goal for contractors during economic downturns. Chinese contractors have creatively attempted to exploit new business areas since they have been confronted with the remarkable decline of market growth in recent years [20]. Thirdly, as in almost all other countries studied, Chinese contractors assign a great deal of importance to their relationships with clients, but they focus more on establishing a long-term friendly relationship (SF22). In the interview at the beginning of this study, it was surprising to discover that occasionally a contractor will bid on an unsuitable project to respond to invitations from target clients, without an intention to win the contract. This phenomenon challenges the argument that successful bidding involves winning a contract instead of merely submitting a bid [38]. Further, improving the experience of a firm's personnel (SF24) also contributes to a contractor's sustainable development. Nevertheless, this factor was ranked relatively low on the list of factors, perhaps because each project can somewhat meet their organizational goals, but the degree of satisfaction may vary. In the case of intense competition during "bad" years, not only do contractors pursue short-term profitability, but they also consider long-term development goals. This strategy allows for contractors to maintain sustainable development in tough times. Moreover, it is interesting to note that these organizational goals may contain more than four factors, which should be tailored to market changes and firm development by individual contractors.

### 5.3. Cluster 4: Organizational Resources

The five factors in cluster 4 are related to organizational resources. Organizational resources are the basis for the development of competitive advantage [39]. In the construction industry, organizational resources are considered fundamental for contractors to acquire and execute their contracts. Moreover, contractors should have abundant resources at hand, including experienced personnel (SF20, SF21, and SF23) and capital for new projects (SF27). Nevertheless, a new project ought to agree with a

firm's business strategy (SF19) so that there is justification to commit one's organizational resources. Having a resource advantage is conducive to winning a contract and achieving its expected benefits in the project execution phase. Bidding is also a process for contractors to plan their resources efficiently. Contractors may lose in the bidding process if their resources are not competitive or adequate, or if their resources are not planned efficiently in the proposal [40]. Similarly, project execution is also a process through with contractors utilize their resources effectively to achieve their expected benefits. Therefore, a contractor may be more willing to bid on a project and have higher confidence in winning a contract when it has abundant organizational resources. This is echoed by Tan, et al. [41], who suggest that contractors should leverage their specific resources to generate a competitive advantage to win contracts.

### 5.4. Cluster 6: Local Business Environment

Localized management has been proven to be significant in the modern age of the construction industry. The motivations for a contractor to place emphasis on local conditions are multifaceted. First, most of the organizational resources during construction can be acquired in the local market; these resources include labor force, staff, materials, equipment, funding, and subcontractors. Second, an excellent relationship with the local community and government is an essential factor affecting project performance [36]. Finally, it is particularly important to comply with local industry policies/regulations during construction. According to Porte and Kramer [42] and Petrovic-Lazarevic [35], relationships with locals and compliance with local industry policies/regulations are two of the most critical CSR activities contributing to a contractor's sustainable development. However, previous studies have paid little attention to these local conditions.

The local social environment (SF33) is one type of local condition and includes resources and customs. Cross-regional operations account for a significant proportion of contractors' total business in China [20]. Therefore, it is impossible to request all the resources from headquarters. Moreover, these resources are deemed to be fundamental for contractors to win contracts and achieve their expected profits. It is, therefore, necessary to assess the possibility of obtaining competitive resources in the local market before bidding. Another factor falling in the category of the local social environment is customs, such as the corruption of officials and local people's attitudes towards environmental pollution. Understanding local customs will help contractors predict the potential difficulties and additional costs to be encountered in the future construction phase, as well as to develop good relationships with the local government officials and residents around the project. Local industry policy (SF32) is another type of local condition. The construction process and inspection of work are subject to the supervision of the local government, which is responsible for making policies or regulations for the local construction industry. Some policies that favor local business establishments may bring great disadvantages to contractors who originate from other provinces. As a result, contractors are more likely to "hit a brick wall" when rushing to bid, let alone achieve sustainable development during economic downturns.

### 5.5. Cluster 2: Project Requirements

The factors in this cluster are related to project requirements, such as labor force, staff, material, equipment, funding, subcontractors, the management competency of the contractor, etc. Even though this cluster does not rank at the top for determining contractors' BNB decisions in this study, Wanous, Boussabaine, and Lewis [22] noted that it is of paramount importance for contractors to fulfill the bidding conditions required by the client. This requirement exists in both the bidding and construction phases. In the bidding phase, the important requirements include the criteria of the bid selection (SF5), the quality of the bidding documents (SF14), and bid security (SF16), except time for bid preparation (SF18). According to the survey, SF18 is considered unimportant, perhaps because the owner usually gives contractors enough time to prepare a bid. In the construction phase, the required subcontractors (SF2), complexity of the project (SF3), needs of the work (SF7), project duration (SF10), and performance

security (SF16) are straightforward requirements, but the amount of work that the client carries out regularly (SF1) and the influence of the client in making recommendations in the construction market (SF6) implicitly emphasize the requirement for the contractor to have project management competency. Commonly, the higher the value of SF1, the higher the requirements. Moreover, SF6 ranked 31st on the list of factors, which means that SF6 is a relatively unimportant factor. Another relatively unimportant factor is project location (SF11), which ranked 28th on the list of factors, but location can define the surroundings of a project. In both the bidding and construction phases, these requirements may vary depending on the client and the characteristics of the specific project.

### 5.6. Cluster 5: Expected Competition

The factors in cluster 5 indicate the expected competition. Bidding means competing with other contractors. As suggested by the Chinese military strategist, Sun Tzu, a competitive participant should, first of all, obtain an understanding of his or her rivals [43]. In this study, the essential information for understanding one's competitors in BNB decisions encompasses the competence of the expected competitors (SF29), the current workloads of the expected significant competitors (SF30), and the number of expected competitors (SF31). Oo et al. [44] proved that contractors' decisions to bid in Hong Kong and Singapore decreased as the number of competitors (SF31) increased. However, (SF30) is not an important factor according to the survey, while a firm's current workload of projects (SF20) ranked 12th on the list of factors. A suitable explanation is that a competitor's ability to organize resources for a new project is unpredictable. Another factor for predicting the competition is SF28. According to the principles of economics, if there are more projects available for bidding, the competition is relatively less. Despite its significance, expected competition was ranked at the bottom of the list of clusters, perhaps because the contractors have become accustomed to the intense competition in the Chinese construction market.

### 5.7. Further Discussion on the Clusters

The above analysis shows that there are six clusters of essential factors, and each cluster comprises its factors determining contractors' BNB decisions. These six clusters can be further classified into three groups based on their attributes from a contractor's perspectives. Expected profitability and organizational goals can be used to assess the extent to which a new project may meet the contractor's strategy. Assessment of organizational resources, local conditions, and project requirements can help a contractor determine his or her competency for a project. Expected competition is the third category, which provides a list of factors that can be used to examine the degree of competition. As a result, a BNB decision can be broken into a three-step modular decision process, including a contractor's strategies, a contractor's competency for a project, and the expected competition, as shown in Table 5. This three-module decision process can serve as a new perspective for BNB decisions and help senior managers better understand the focus of decision-making.

**Table 5.** Ranks of strategic determinants of BNB decision-making during economic downturns.

| Strategic Determinant | Rank | Cluster | Importance Index |
|---|---|---|---|
| Contractor's strategies | 1 | Cluster 1: expected profitability | 83.45 |
| | | Cluster 3: organizational goals | 75.36 |
| Contractor's competency for a project | 2 | Cluster 4: organizational resources | 70.49 |
| | | Cluster 6: local business environment | 69.29 |
| | | Cluster 2: project requirements | 68.49 |
| Expected competition | 3 | Cluster 5: expected competition | 65.52 |

## 6. Conclusions

The sustainable growth of construction contractors depends heavily on the ability to find suitable projects to bid on, especially in economic downturns. Certainly, an economic entity has many cycles, and contractors have to know how to manage BNB decision factors effectively in economic downturns.

Our research findings indicate that 31 factors are vital to the effectiveness of contractors' BNB decisions during a "bad" economic period; these factors contain six clusters: expected profitability, organizational goals, organizational resources, local conditions, project requirements, and expected competition.

The research findings highlight that contractors' BNB decision factors are related to expected profitability, including the terms of payment, the reputation of the client regarding his commitment to making timely payments, the original prices estimated by the client, a profit track record for similar projects, and the contract type. Moreover, a BNB decision-making process can be divided into three modules: a contractor's strategies, a contractor's competency for a project, and expected competition.

These research findings echo previous studies in that a contractor's BNB decision-making process is complicated and subject to a variety of factors. There are no exceptions to these factors in economic downturns. Although the data were collected from the Chinese construction industry, the determinants of BNB decisions are still relevant for contractors in other countries to select proper projects to bid on to realize their sustained development goals during economic downturns.

**Author Contributions:** Conceptualization, K.Y. and J.W.; supervision, L.W.; writing—original draft preparation, J.W.; writing—review and editing, K.Y. and Y.S. All authors have read and agreed to the published version of the manuscript.

**Funding:** This work has been supported by Fundamental Research Funds for the Central Universities of China (2019CDSKXYJSG0041) and National Natural Science Foundation of China (71871033).

**Conflicts of Interest:** The authors declare no conflict of interest.

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
