# Peer review of "Will Bid/No-Bid Decision Factors for Construction Projects Be Different in Economic Downturns? A Chinese Study"

_applsci, doi:10.3390/app10051899_

Round 1

Reviewer 1 Report

Thank you for giving me the opportunity to read your interesting paper on bidding decisions in the construction industry!

Title: Please consider if you want to add the construction sector in your title because currently the reader doesn’t know who bids on what.

Title: Please also consider if you want to add China as your regional focus.

Introduction: Please add the contributions of your research and provide more infor

Methodology: Please provide more information on the interviews (3.1): Demographics of the respondends? Wehen were the interviews conducted? What type of questioning method? Where the interviews transcribed? How were the texts analyzed and the results derived?

Conclusion: Please provide a conclusion with (1) a short summary of your findings, (2) limitations, (3) future research.

References: You preferably cite somewhat old publications. Please add more recent publications.

Author Response

The authors would like to thank the editors and reviewers for their time in effort in reviewing our manuscript. We hope the changes listed below have made the manuscript suitable for publication and we look forward to your response.

Point 1: Title: Please consider if you want to add the construction sector in your title because currently the reader doesn’t know who bids on what.

Response 1:

The authors would like to thank the reviewer for the suggestion. The revised title is “Will Bid/No-bid Decision Factors for Construction Projects be Different in Economic Downturns? A China Study”.

Point 2: Title: Please also consider if you want to add China as your regional focus.

Response 2:

The authors would like to thank the reviewer for the advice. Because of the research material from China’s construction industry, we have added China as our regional focus in title.

Point 3: Introduction: Please add the contributions of your research and provide more infor

Response 3:

The authors would like to thank the reviewer for the suggestion. We have revised the introduction. Please see the attachment.

Point 4: Methodology: Please provide more information on the interviews (3.1): Demographics of the respondents? When were the interviews conducted? What type of questioning method? Where the interviews transcribed? How were the texts analyzed and the results derived?

Response 4:

The authors would like to thank the reviewer for the suggestion. The interview with professionals was conducted to refine the preliminary factors from literature review. We have revised the content of the research method in the manuscript to achieve a clear expression. Please see the attachment.

Point 5: Conclusion: Please provide a conclusion with (1) a short summary of your findings, (2) limitations, (3) future research

Response 5:

The authors would like to thank the reviewer for the suggestion. We have added the conclusion in the manuscript. Please see the attachment.

Point 6: References: You preferably cite somewhat old publications. Please add more recent publications.

Response 6:

The authors would like to thank the reviewer for the suggestion. We have added recent publications as follow.

  1. Yan, P.; Liu, J.; Skitmore, M. Individual, group, and organizational factors affecting group bidding decisions for construction projects. Advances in Civil Engineering 2018, 2018, 1-10
  2. Pettinger, T. Causes of business cycle. Availabe online: https://www.economicshelp.org/blog/386/economics/causes-of-business-cycle/#more-386 (accessed on 15 February 2020).
  3. Pettinger, T. Economic booms. Availabe online: https://www.economicshelp.org/blog/glossary/booms/ (accessed on 15 February 2020).
  4. Pettinger, T. The accelerator effect. Availabe online: https://www.economicshelp.org/blog/glossary/accelerator-effect/ (accessed on 15 February 2020).
  5. National data. Availabe online: http://data.stats.gov.cn/easyquery.htm?cn=C01 (accessed on 15 February 2020).
  6. China's GDP is nearly RMB 100 trillion and the growth rate is 6.1% in 2019. Availabe online: http://www.gov.cn/shuju/2020-01/18/content_5470531.htm (accessed on 15 February 2020).
  7. Pettinger, T. Economic downturn definition. Availabe online: https://www.economicshelp.org/blog/6976/economics/economic-downturn-definition/ (accessed on 15 February 2020).
  8. Wu, S.; Pan, F. Statistical analysis of SPSS, 1st ed.; Tsinghua University Press: Beijing China, 2014; pp. 315-330.
  9. Porter, M.E. Competitive advantage, 1st ed.; Citic Press: Beijing China, 2014; pp. 29-51.

Reviewer 2 Report

Fig.1:

102 The factors are divided  into positive and negative factors to represent those factors that encourage contractors to bid or not to bid respectively.

But see Fig 1 - why all the factors are connected by arrows with both positive and negative biddings factors ?

Table 3.

Where is the boundary between the categories of factors?

Table 5:

where are boundaries between Basic categories and  boundaries between clusters

Fig 2,3,4

which means the horizontal scale in figures 2, 3, 4 ?

The number of clasters in Figure 2 is not justified. Maybe there are 3 clusters here?

Author Response

The authors would like to thank the editors and reviewers for their time in effort in reviewing our manuscript. We hope the changes listed below have made the manuscript suitable for publication and we look forward to your response.

Point 1: Fig.1:102 The factors are divided into positive and negative factors to represent those factors that encourage contractors to bid or not to bid respectively. But see Fig 1 - why all the factors are connected by arrows with both positive and negative biddings factors?

Response 1:

The authors would like to thank the reviewer for the comments. We have removed the Fig 1 in the revised literature review because it is useless. Please see the attachment.

Point 2: Table 3. Where is the boundary between the categories of factors?

Response 2:

The authors would like to thank the reviewer for the comment. We have revised the table. Please see the attachment.

Point 3: Table 5: where are boundaries between Basic categories and boundaries between clusters?

Response 3:

The authors would like to thank the reviewer for the comment. We have revised the table. Please see the attachment.

Point 4: Fig 2,3,4

which means the horizontal scale in figures 2, 3, 4 ?

The number of clusters in Figure 2 is not justified. Maybe there are 3 clusters here?

Response 4:

The authors would like to thank the reviewer for the comments and suggestion.

The horizontal scale represents the Euclidean distance from a variable to the center point of the cluster. We think there are cluster 1 and cluster 2 in the first hierarchy. The attributes of factors in cluster 2 are similar, so no further cluster is required.

Reviewer 3 Report

I have reviewed the manuscript and come up with the following feedback:

Introduction:

The narrative needs to be improved to ensure the readability of the content.  The problem statement is too weak; therefore, it has to be supported and justified with stronger evidences

Literature review:

Too weak and looks like a "Jim did that Jack did that" approach There should be clear logical links and various strands of thought should be clearly identified. It is important to position your research appropriately within this literature review. The contribution of your work to our collective understanding will be clearer if you can make the connections.

Research Methods:

The factors were identified through interviews with experts in China. Who are these experts? Demographics of the respondents must be provided with a clear justification of their suitability and quality of their responses.

Conclusion:

The conclusion part is missing. The authors have to provide the concluding remarks of the research.

Based on my review, I recommend the manuscript to undergo major revision.

Author Response

The authors would like to thank the editors and reviewers for their time in effort in reviewing our manuscript. We hope the changes listed below have made the manuscript suitable for publication and we look forward to your response.

Point 1: Introduction: The narrative needs to be improved to ensure the readability of the content. The problem statement is too weak; therefore, it has to be supported and justified with stronger evidences.

Response 1:

The authors would like to thank the reviewer for the suggestion. We have revised the introduction. Please see the attachment.

Point 2: Literature review: Too weak and looks like a "Jim did that Jack did that" approach There should be clear logical links and various strands of thought should be clearly identified. It is important to position your research appropriately within this literature review. The contribution of your work to our collective understanding will be clearer if you can make the connections.

Response 2:

The authors would like to thank the reviewer for the suggestion. We have revised the literature review. Please see the attachment.

Point 3: Research Methods: The factors were identified through interviews with experts in China. Who are these experts? Demographics of the respondents must be provided with a clear justification of their suitability and quality of their responses.

Response 3:

The authors would like to thank the reviewer for the suggestion. The interview with professionals was conducted to refine the preliminary factors from literature review. We have revised the content of the research method in the manuscript to achieve a clear expression. Please see the attachment.

Point 4: Conclusion: The conclusion part is missing. The authors have to provide the concluding remarks of the research. Based on my review, I recommend the manuscript to undergo major revision.

Response 4:

The authors would like to thank the reviewer for the suggestion. We have added the conclusion in the manuscript. Please see the attachment.

Round 2

Reviewer 1 Report

Thank you very much for this very good revision!

Author Response

Point 2: Thank you very much for this very good revision!

Response 2:The authors would like to thank the reviewer for the comment.

Reviewer 3 Report

The authors have addressed the issues raised in the first round of review process. However, the manuscript could be proofread by the professional to improve the readability.

Author Response

Point 1:The authors have addressed the issues raised in the first round of review process. However, the manuscript could be proofread by the professional to improve the readability.

Response 1:The authors would like to thank the reviewer for the comment and suggestion. We submit the manuscript to MDPI for editing.Please see the attachment.